# Securing Non-Terrestrial FSO Link with Public Key Encryption against Flying Object Attacks

**Daniel Hicks** [1]**, Fatma Benkhelifa** [2] **, Zahir Ahmad** [1]**, Thomas Statheros** [3] **, Osama Saied** [4]**,
Omprakash Kaiwartya** [4] **and Farah Mahdi Alsallami** [1,*]

[1] The Faculty of Engineering, Environment and Computing, Coventry University, Coventry CV1 5FB, UK; hicksd2@uni.coventry.ac.uk (D.H.); ad7175@coventry.ac.uk (Z.A.)

[2] School of Electronic Engineering and Computer Science, Queen Mary University of London, London E1 4NS, UK; fatma.benkhelifa@kaust.edu.sa

[3] The Centre for Future Transport and Cities, Coventry University, Coventry CV1 2TE, UK; ac5304@coventry.ac.uk

[4] Department of Computer Science, Nottingham Trent University, Clifton Campus, Nottingham NG11 8NS, UK; osama.saied@ntu.ac.uk (O.S.); omprakash.kaiwartya@ntu.ed.ac.uk (O.K.)

\* Correspondence: ad9051@coventry.ac.uk

**Abstract:** Free Space Optical (FSO) communication has potential terrestrial and non-terrestrial applications. It allows large bandwidth for higher data transfer capacity. Due to its high directivity, it has a potential security advantage over traditional radio frequency (RF) communications. However, eavesdropping attacks are still possible in long non-terrestrial transmission FSO links, where the geometry of the link allows foreign flying objects such as Unmanned Aerial vehicles (UAVs) and drones to interrupt the links. This exposes non-terrestrial FSO links to adversary security attacks. Hence, data security techniques implementation is required to achieve immune FSO communication links. Unlike the commonly proposed physical layer security techniques, this paper presents a lab-based demonstration of a secured FSO communication link based on data cryptography using the GNU Radio platform and software-defined radio (SDR) hardware. The utilized encryption algorithm (Xsalsa20) in this paper requires high-time complexity to be broken by power-limited flying objects that interrupt the FSO beam. The results show that implementing cryptographic encryption techniques into FSO systems provided resilience against eavesdropping attacks and preserved data security. The experiment results show that, at a distance of 250 mm and laser output power of 10 mW, the system achieves a packet delivery rate of 92% and transmission rate of 10 Mbit/s. This is because the SDR used in this experiment requires a minimum received electrical amplitude of 27.5 mV to process the received signal. Long distance and higher data rates can be achieved using less sensitive SDR hardware.

**Keywords:** data security; free space optical communication; UAV; non-terrestrial communication; public key encryption

## 1. Introduction

Free space optical (FSO) communication is one of the emerging breakthroughs to support the 6G networks. FSO links promise high data rates, a licence-free spectrum, and massive connectivity. This technology supports fixed terrestrial point-to-point communication for military applications, mobile communications, and internet service providers [1]. This system has also been proposed to provide non-terrestrial communication using satellite and unmanned aerial vehicles (UAVs) networks [2–4].

With any mode of FSO communication, terrestrial or non-terrestrial, it is crucial that sensitive information is kept secure. Due to the directional nature of optical beams, FSO is believed to offer superior security to radio frequency (RF) that makes it difficult to intercept [5,6]. For this reason, the literature related to physical layer security in wireless

optical communications is scarce [5]. Although it is true that FSO offers more robust physical layer security than traditional RF transmissions, it is not a strong enough argument to disregard data security [5]. The study in [6] delves into physical layer security in FSO for the "difficulty of breach by a third party" compared to cryptographic techniques. Whilst this is true, it is also the case that physical layer encryption techniques often require additional hardware devices which drive the costs compared to mathematical cryptographic techniques at the presentation layer. To the best of our knowledge, none of the previous studies investigated this type of cryptographic encryption in FSO but instead focused on the physical layer.

Considering security is paramount to communication systems, the literature on the presentation layer security is plentiful. Bernstein et al. [7] highlights some of the underlying problems with cryptographic libraries such as OpenSSL and addressed them with a new library called Networking and Cryptography library (NaCl) (also known as salt). This library has some core features such as "No data flow from secrets to load addresses" and "centralizing randomness" to achieve higher performance and security. In [8], salt password hashing was used to secure data storage and transmission over a cloud computing network. The "salt" represents a random string which is hashed and combined with a hashed private key. The combination is once again hashed to guarantee the data cannot be decrypted under any condition. Similarly, the work in [9] used salt cryptography to secure data transmissions by embedding the transmitted information into a video with promising results for how robust it is to attacks.

Likewise, more advanced data security techniques, such as elliptic curve cryptography, exist. In [10] the performance advantages of elliptic curve cryptography (ECC) were compared to other public key systems such as Rivest–Shamir–Adleman (RSA) or Diffie–Helman. The study concluded that the reason for ECC's success was due to industrial adaptation, which is as important as the performance advantages of a data security proposal. Recently, a non-terrestrial satellite or UAV-based FSO system whilst employing quantum key distribution (QKD) to secure the transmission was proposed in [11,12]. However, quantum-based security is cost ineffective because the technology is still in its infancy.

A balance of security and performance is important for adopting technologies such as FSO. It has been shown that strong upper-layer encryption techniques already exist and have been used to ensure secure data transmissions; however, they are not usually applied to a communication system using an FSO channel.

*Motivation and Original Contributions*

The most discussed advantage of FSO communication is that it allows huge bandwidths and data capacity, as demand is always growing in both the industrial and commercial sectors. However, eavesdropping attacks are possible when the transmission distance is large. For this reason, securing FSO links is essential to preserve the security of data and help with the adoption of new FSO technologies. Therefore, the main original contributions of this study are

- Implement a secure FSO link using the NaCl library to generate encryption keys in software using GNU Radio Companion (GRC) 3.7.13.5 and investigate its performance in simulation. In particular, encryption algorithm (Xsalsa20) is used due to its high time complexity. Hence, it provides sufficient data security against adversary UAV that has limited computing power.
- Demonstrate a laboratory-based experiment of the secure FSO link using optics and software-defined radio (SDR) transceiver and compare results to the simulation.

The rest of the paper is organized as follows. The proposed system model is described in Section 3. Secure FSO link performance in GNU Radio simulation is discussed in Section 4. The experimental demonstration of the link is given in Section 5. Finally, conclusions are provided in Section 6.

## 2. System Model

FSO is known for its narrow optical beam which offers immunity against security attacks. However, the beam spreads out with propagation distance, $z$, as follows [13,14]:

$$\omega(z) = \omega_\circ \sqrt{1 + \left(\frac{\lambda z}{\pi \omega_\circ^2}\right)^2} \tag{1}$$

where $\omega_\circ$ and $\lambda$ are the beam waist and wavelength, respectively, of the laser at the transmitter.

Figure 1 shows (a) schematic and (b) geometry of a security attack scenario on a non-terrestrial FSO link between a satellite and ground station as the transmitter (Alice) and receiver (Bob), respectively. The attacker (Eve) is a UAV that interrupted the optical beam. The work in [15] showed that an FSO beam has a spread diameter $d \approx D\theta + R$ of 50 cm for a divergence angle of $\theta = 0.1$ mrad and link length of $D = 5$ km, where $R$ is the beam diameter at the transmitter. In satellite FSO communications, when the link length ranges from 500 km for low earth orbit satellites, to 500 million km for deep space optical links, the beam radius expands between 6.63 m to $2.19 \times 10^5$ m ([16], p. 164). This beam diameter expansion is 1000 times less than RF-based satellite communications beam [17,18] and it can be controlled using optics [19,20]. However, very narrow beam divergence is not desirable because it causes mis-alignment errors due to satellite vibration or platform jitter [17]. With the advances in UAV technologies, a flying attacker of 10 cm receiver aperture can interrupt the broad optical beam and align to the transmitter which exposes the FSO link to security threats [21], as illustrated in Figure 1b. In the terrestrial FSO case, Eve was assumed to be a sufficiently sensitive device that can collect a fraction of leak power $< 10^{-2}$, otherwise, it causes a power reduction that notifies the legitimate peers Alice and Bob [5]. The study in [5] also showed that relying on the physical layer security of the FSO is not sufficient when a fraction of leak power $> 10^{-2}$. Hence, an upper-layer data encryption technique is required.

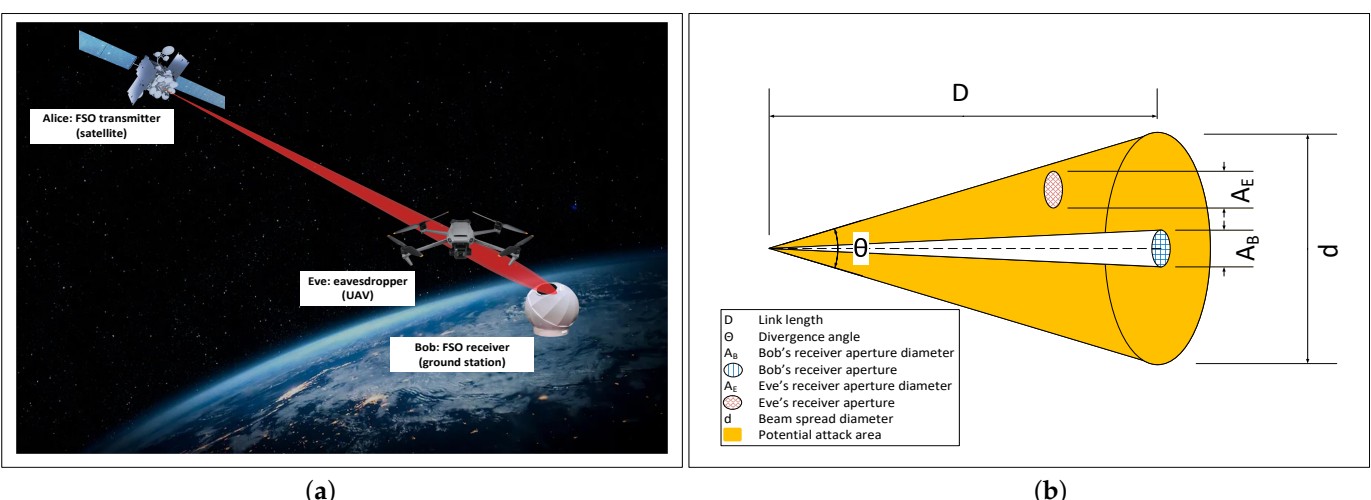

(**a**)    (**b**)

**Figure 1.** Security attack on satellite to ground-station FSO link by a UAV that interrupts the optical beam: (**a**) schematic (not to scale) and (**b**) geometry of the link.

In this study, we propose using a presentation layer data encryption technology to secure the transmission of non-terrestrial FSO communication links. Figure 2 illustrates a block diagram of the secure FSO system under investigation. The system is made in a simulation using the GNU Radio platform and as a prototype using an SDR transceiver and optical hardware. GNU Radio provides the user interface and handles the data processing for the transmitter and receiver in the background. The transmitter end allows a user to enter data to be encoded, encrypted, modulated, and shape burst. At the receiver end, the incoming data will be filtered, demodulated, decrypted, and decoded.

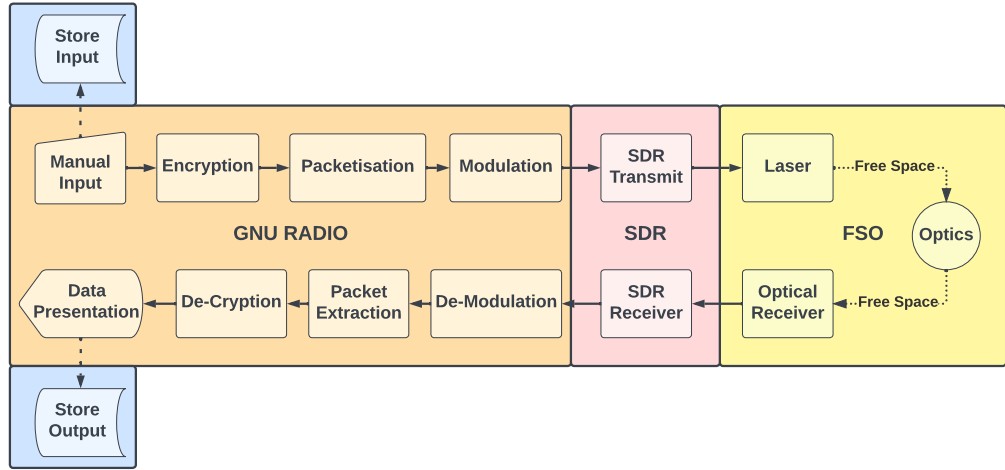

**Figure 2.** System block diagram.

The GNU-Radio module, responsible for implementing the encryption techniques, is an out-of-tree module called gr-nacl, developed by Wunsch et al. [22] . This module uses a well-known library called NaCl that provides functions for high-speed network communication, encryption, and signatures [7,22]. This encryption technique has continuously proven to be secure despite advancements in modern computational power. The most efficient attack on the encryption algorithm (Xsalsa20), which is used in NaCl to generate encryption keys, showed that this technique only breaks 8 of 20 rounds of encryption with time complexity of $2^{250}$ [23]. This provides sufficient data security against power-limited eavesdroppers [24]. The NaCl library was implemented into this GNU Radio module using another library called libsodium [25]. It aims to wrap all the complex NaCl functions into simple high-speed calling functions. Finally, gr-nacl puts these functions into GNU Radio blocks that can be integrated with the rest of the workspace.

The NaCl cryptography library contains an abundance of algorithms and functions. However, only a handful of these have been implemented into gr-nacl: key generation, public encryption, private encryption, and stream encryption. We implemented public encryption techniques using the public encrypt/decrypt blocks and keypair generation blocks for both the sender and recipient.

Public encryption is used to avoid the need to exchange encryption keys across a secure channel. Moreover, this technique allows two parties to establish a secure channel by exchanging encryption keys across an insecure channel which is the case in non-terrestrial FSO.

## 3. Secure FSO Link in GNU Radio Simulation

The implementation of the secure FSO link in the GNU Radio platform is split into four main parts: the initial setup, the transmitter, the FSO channel, and finally the receiver. Each part is explained as follows.

### 3.1. Initialisation

Before any communication can occur, the initial setup must take place; this includes setting variables in GNU Radio for system parameters, as illustrated in Figure 3. There are also QT graphical user interface (GUI) blocks that control the user interface (UI) elements of the flowgraph. The QT GUI Range blocks allow the user to change the value of certain aspects of the FSO link using a slider and the QT GUI Tab widgets allow the user to switch between tabs for different system performance views.

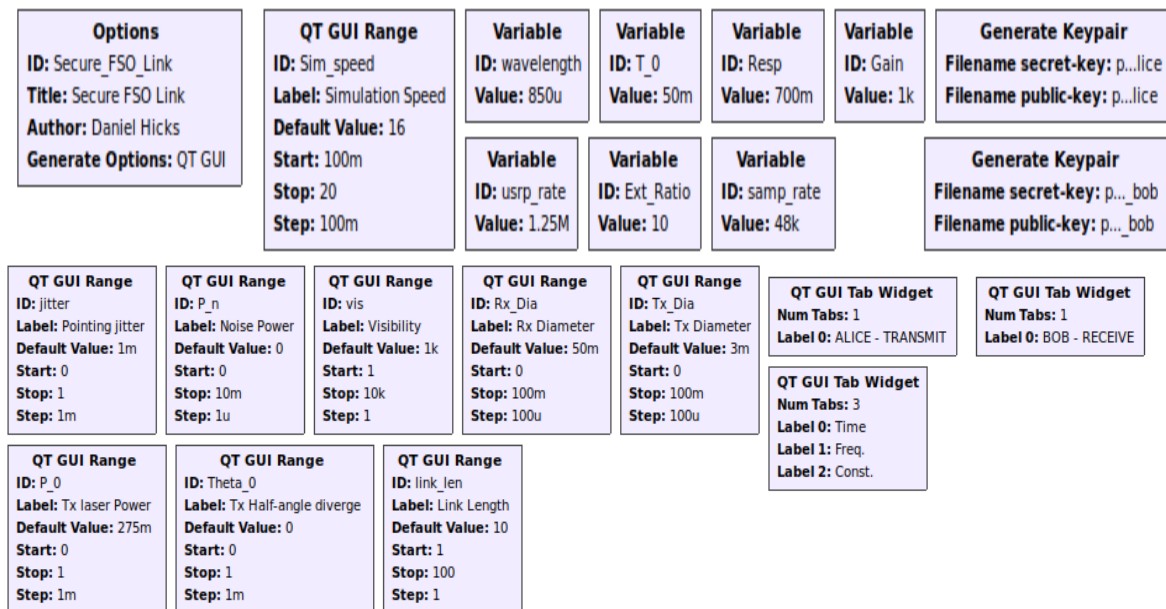

**Figure 3.** Secure FSO Initialisation blocks in GNU Radio.

At this stage, the encryption keys are generated: one pair for each transmitter and receiver. The Generate Keypair block uses a NaCl function called crypto_box_keypair to generate a 32-byte secret key and a corresponding 32-byte public key. These random keys are generated using the Curve25519 function in NaCl which is a high-speed key generation method resulting in smaller keys than other cryptography methods such as RSA. The function first generates a random 32-byte integer which is considered to be a private key, then computes a corresponding public key. The properties of elliptic curves are used to combine a base point with the random private key to produce the public key. Here, the curve used in Curve15519 is always defined as $y^2 = x^3 + 486662x^2 + x$. This public key is created so that it is impossible to decipher the private key from the public key and base point without considerable computational effort.

For encryption to occur, the sender must have the recipient's public key and vice versa. Therefore, the system becomes more secure as an attacker needs to know both the sender and recipient's public keys, which means the public keys can be exchanged across an insecure channel. In the case of this simulation, it is important to point out that this key exchange is assumed to have already taken place because the keys are stored on the same local drive. In the real world, the keys would have to be generated and then the public keys need to be exchanged for any secure communication to occur.

### 3.2. Transmitter

Once the initialisation stages are complete, the transmitter can send secured messages. There are six main stages that the transmitter goes through to prepare the message to be sent through free space, as depicted in Figure 4.

The first stage is for the user to enter a message into a UI entry box. Next, the message is passed to the custom message handler and the UI entry box is cleared; this message handler is coded specifically for this system. The message handler prepares the message format for encryption, clears the UI entry box, prints to the terminal, and stores the message in a log.

Therefore, GNU Radio passes asynchronous messages using streams. When the data represent a message type (shown by blocks with grey inputs/outputs), the transferred data will be a polymorphic type (PMT). PMTs are commonly used for asynchronous communications due to their flexibility. The incoming message is a PMT and it is converted to a PMT vector of two elements. The first element is the tag 'msg_clear'; this is needed

as the encryption block searches for PMT vectors with the tag 'msg_clear' so that it can encrypt the second element of the vector, which contains the u8vector data of the message and is coupled with the metadata of its length. The message must be in this format for the encryption blocks to recognise the data and perform accurate encryption. The message is also stored in a log file and written to the terminal as part of the UI to debug the data flow.

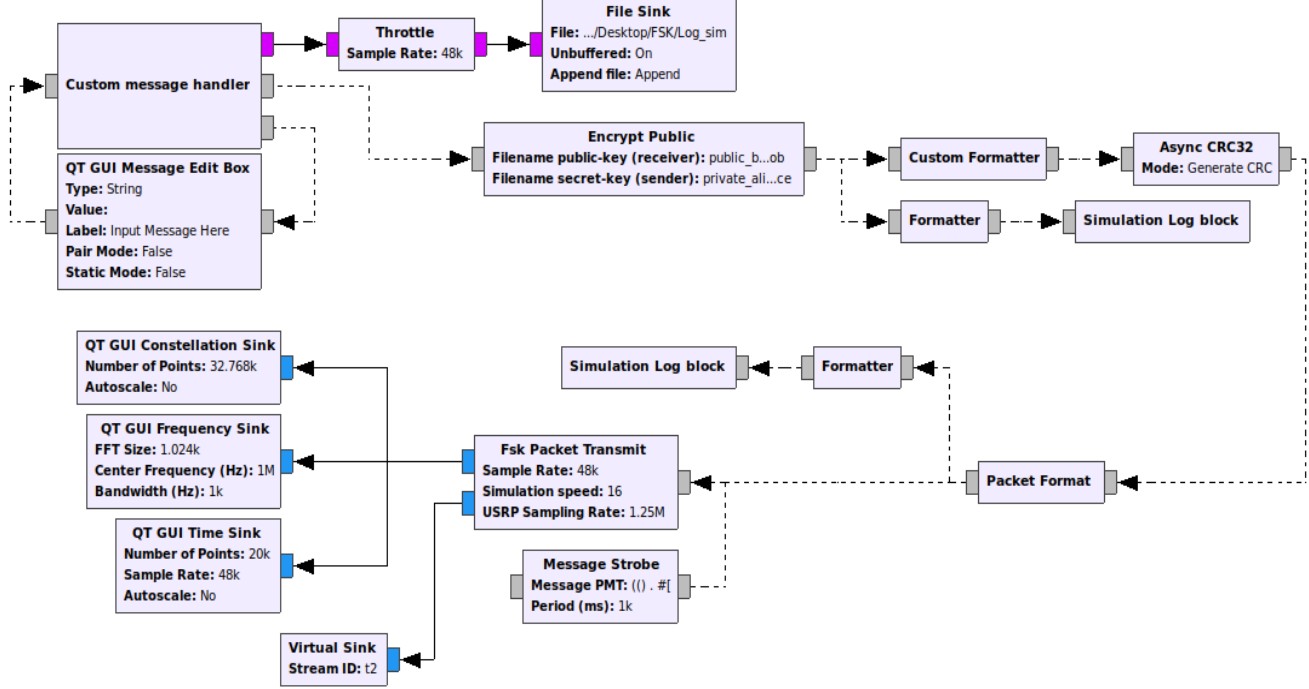

**Figure 4.** Secure FSO link Transmitter blocks in GNU Radio.

Now the message is ready to be encrypted. The libsodium library provides a high-level application programming interface (API) for encryption that will produce an authenticated and encrypted message in one function. The function used by gr-nacl public encryption is crypto_box_easy, which takes a message, a random nonce, the sender's secret key and the recipient's public key to produce an encrypted message bundled with an authentication tag which ensures the message has not been tampered during transmission. The type of encryption used is called Xsalsa20 and the authentication used is Poly1305 [7].

The message is passed through another custom formatter block to change the message type to a PMT construct, containing the nonce and packet in u8vector form. Simultaneously the message is saved to the log and printed to the terminal by the formatter and subsequent simulation log block. Next, a 32-bit CRC is calculated and appended to the end of the message, which further ensures the integrity of the message at the receiver. One more custom formatter block applies a custom packet format that wraps the payload with four bytes of pre-amble, a four-byte access code, and finally four header bytes. For this work, the preamble is equal to $0 \times 55\, 0 \times 55\, 0 \times 55\, 0 \times 55$ in hex or 85 85 85 85 in decimal and the access code is $0 \times E1\, 0 \times 5A\, 0 \times E8\, 0 \times 93$ in hex or 225 90 232 147 in decimal. The final message configuration is ready to be sent to the transmitter for modulation, so once again it is saved to the log and written to the terminal. (Note that a message strobe block is also added here to keep the simulation active and push messages through certain buffers).

This message is passed to the frequency shift keying (FSK) packet transmit a hierarchical block of the gr-control out-of-tree (OOT) [26] module but has been modified to suit the proposed system. A complex voltage control oscillator (VCO) is used and a fractional resampler has been added to control the speed of the simulation by changing the resampling radio.

Now that the data have been modulated, they can be transmitted through the channel. Constellation, frequency, and time sinks are added after modulation as GUI elements to help visualise the signal and measure signal strength.

### 3.3. Channel

FSO systems are prone to random channels, and hence, received power varies due to atmospheric conditions such as fog and turbulence losses [27]. In addition, other sources of randomness occur due to link geometry, such as pointing errors and geometric losses. This can become problematic for the receiver. To simulate an accurate free space optical channel, an OOT module in [28] was used. The FSO channel is implemented using hierarchal blocks as shown in Figure 5. The FSO channel blocks account for channel variables that can be adjusted and tested in simulation.

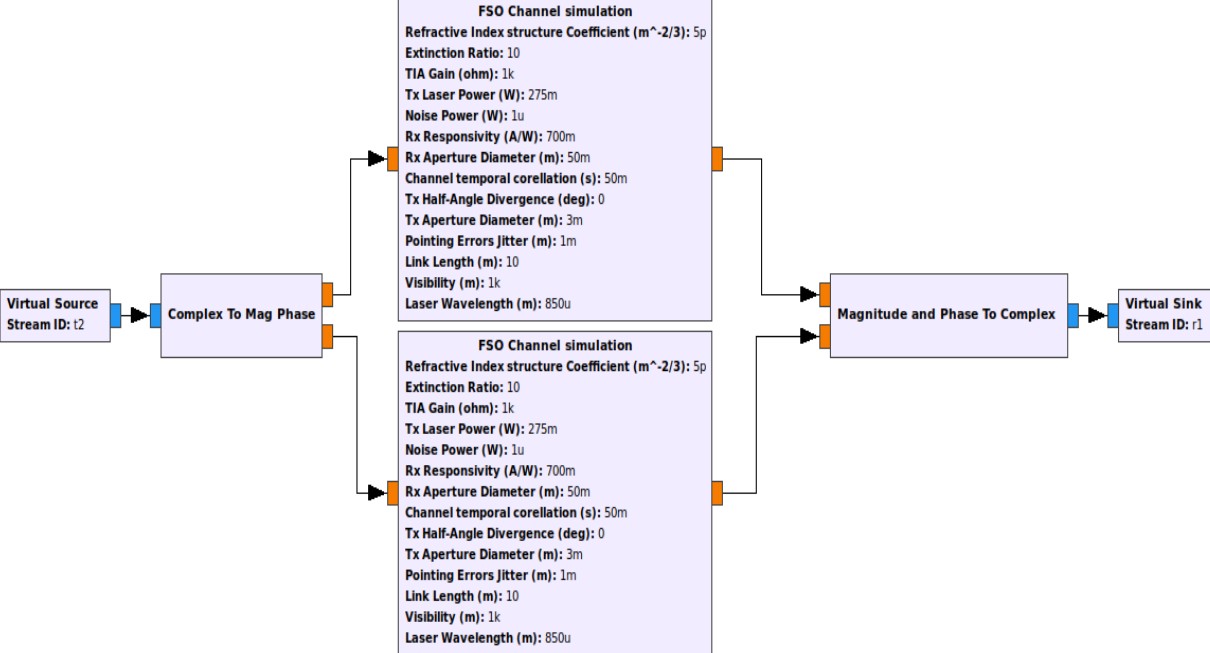

**Figure 5.** Secure FSO link Channel blocks in GNU Radio.

The output of the packet transmit block is complex. However, FSO channel only allows real float vectors to pass through. The data type conversion methods in GNU Radio do not support direct complex-to-float conversion. Alternatively, to deal with this, the transmitted data must be split into its individual magnitude and phase samples and then passed through identical channels before being combined back into a single complex sample.

There are a total of eight channel variables to simulate four types of losses: geometric, turbulence, pointing errors and weather. Geometric losses describe the beam spreads over a distance as in (1); hence, not all the light from the transmitter can be focused onto the receiver. The amount of power lost from this is affected by the transmitter/receiver diameters, the link length and the transmitter half-angle divergence [19]. A higher divergence angle means the light spreads out more, and the receiver does not receive the full power. As the link length increases, the beam will also spread out more, meaning the receiver power will be lower [20].

Turbulence losses are a type of optical loss, caused by slight fluctuations in humidity, pressure and temperature in the air [29]. These losses are negligible in a lab environment but can have a significant impact over a longer distance in free space. For that reason, the simulated turbulence losses are also assumed to be small. The variables that will change

the amount of loss are the refractive index and the channel temporal correlation; however, increasing the link length will also increase losses.

Pointing errors are affected by the alignment between the transmitter and receiver. This type of error is caused by the Jitter variable, which determines how far the centre of the beam can deviate from the centre of the receiver. It is caused by mechanical vibrations or a swaying structure [29]. Other factors, such as the link length, transmitter/receiver diameter and divergence angle, will change how much the jitter affects the final signal strength. For example, a longer link length will amplify these jitters at the receiver. Pointing errors are also negligible in a lab environment, so the pointing error jitter is kept at 0 in the simulation.

Weather losses, or fog losses, refer to losses caused by microscopic particles in the air (usually water droplets) that can reflect or block small parts of the beam and reduce the overall received power at the receiver. These losses are amplified over larger distances and are usually a major hurdle in industrial applications where air quality is lower [30]. However, in a lab, these losses are much less of a problem. The visibility variable has the utmost effect on the weather losses; however, as with other losses, the link length also changes the weather losses.

Each of these variables can be adjusted in the UI while the simulation is running by using a slider to make testing and data gathering as efficient as possible. Now that the data have passed through the simulated channel, depending on the values of the variables, noise is introduced, and the transmitted optical power decays. The receiver recovers the data from the noisy signal so that it can demodulate and decipher the important data from the packet.

*3.4. Receiver*

The receiver has the same stages as the transmitter but in reverse, as given in Figure 6. The data coming into the receiver must first pass through the FSK Packet Receive block before being depacketised and decrypted.

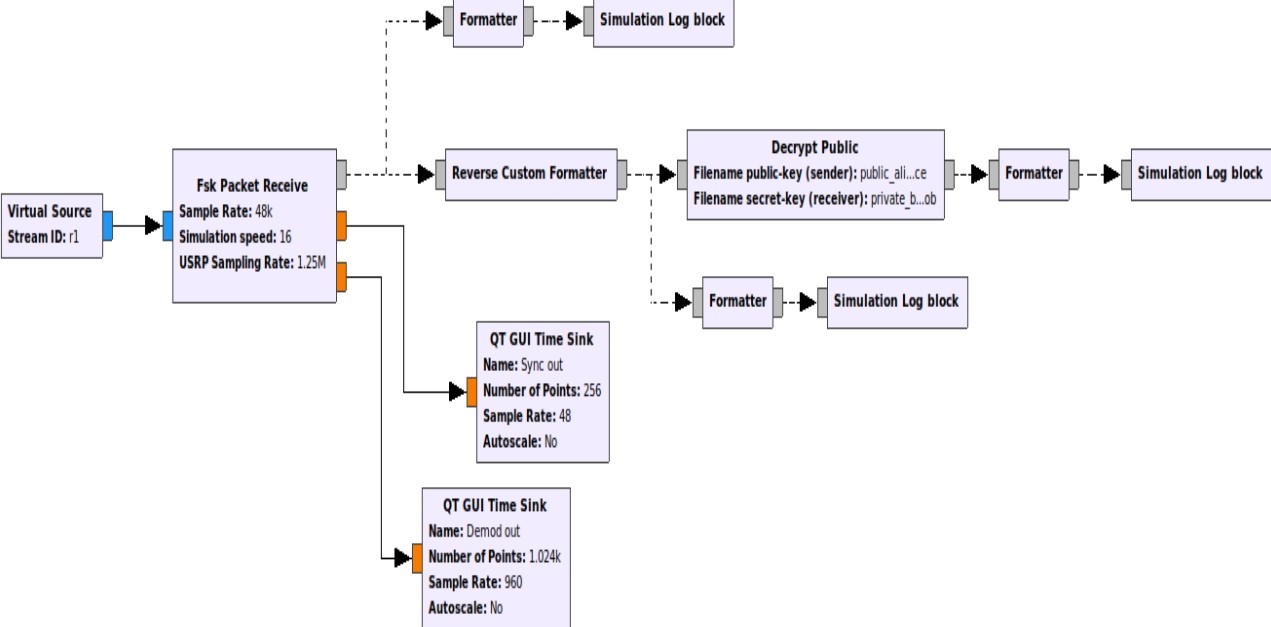

**Figure 6.** Secure FSO link Receiver blocks in GNU Radio.

As explained in Section 3.2, the access code is part of the header that consists of four bytes. The FSK Packet Receive block locates and discards the access code and any data before it (this includes the preamble). Then, it passes the rest of the binary data to the next block. Lastly, the bits are repacked into bytes and then the CRC is verified and passed on. Now the data will consist of just the nonce data and the encrypted packet data.

To extract the packet data and prepare the data for decryption, a reverse custom formatter was created. The reverse custom formatter will add the tag 'nonce' to the nonce data and pack it into a PMT list, and the encrypted message is assigned the tag 'msg_encrypted' and is also packed into a PMT list, this is so that the decryption block locates the nonce and the encrypted data.

Next, the message is verified and decrypted by the Decrypt Public block from gr-nacl using the crypto_box_open_easy function from libsodium. If the message is properly decrypted, then the message is sent to the formatter to be written to the log, and it is also written to the terminal. If the message is not decrypted, then the block returns a message to indicate that the message cannot be decrypted and provide a reason, e.g., no nonce detected or the encryption key was invalid. In the last stage, the message is logged before the reverse formatter, after the reverse formatter and finally, after the decryption.

### 3.5. Simulation Results

The communication link in the simulation has been designed to replicate a realistic lab environment. The simulation parameters of the link are link length = 0.5 m, receiver diameter = 4 mm, transmitter diameter = 3 mm and transmitter laser power = 10 mW. The laser's half-angle divergence is assumed to be very small (very close to 0) due to the short link length and use of lenses that collimates the beam. System performance is measured by packet delivery rate (PDR) which is the ratio of the received packet number to the transmitted packet number.

Figure 7 shows the FSO link output of a secure conversation between Alice and Bob. The output shows a different format of the encrypted and packetized message at Alice and Bob. For example, the difference between an encrypted message and an encrypted packet is that the nonce and message have been combined into one and at the start of the message the four-byte preamble is added (85 85 85 85), four-byte sync is added (225 90 232 147), and four header bytes are added.

In an eavesdropping attack, Eve can receive the data by using the same demodulation as Bob. Eve can also generate their keypair using the same method as Alice and Bob to decrypt the incoming messages using their private key and the sender's (Alice) public key. However, since Eve does not have the correct private encryption key, they cannot decrypt the gathered data.

Figure 8 shows that Eve has successfully received the message and can decipher the packet format. Eve can also identify the nonce and message begin but they cannot decrypt the message. The crypto_box_open_easy function from libsodium returns −1 "Failed to decrypt message." meaning the message cannot be decrypted and remains a secret.

To study the performance of the proposed system under realistic satellite to ground-station beam geometry and channel conditions, a simulation-based FSO link is implemented with realistic parameters shown in Table 1 for low earth orbit satellite mission that uses miniature optical communication transceiver (MOCT) developed at the University of Florida [16]. Figure 9a shows that this FSO link achieves a PDR of 88% using transmitted laser power of 10 W. When the transmitted laser power is reduced to 0.1 W (i.e., a fraction of leakage power $10^{-2}$ [5]), the PDR decreases to 0%, unless Eve is 100 times more sensitive or can boost the received signal using an amplifier with gain of 20 dB. In this case, the PDR increases to 96% as illustrated in Figure 9b. The results also shows that Eve failed to decrypt any encrypted messages.

**Figure 7.** FSO link output of a secure conversation between Alice and Bob under ideal channel conditions.

**Figure 8.** FSO link output of Eve intercepting a message sent from Alice under ideal channel conditions.

(a)

(b)

**Figure 9.** Simulation results of realistic satellite to ground-station FSO link parameters when the transmitter laser power values are: (**a**) 10 W and (**b**) 0.1 W and the receiver uses 20 dB amplifier.

**Table 1.** Satellite to ground-station link parameters for low earth orbit satellite mission that uses miniature optical communication transceiver (MOCT) developed at the University of Florida [16].

| Parameter | Value |
| --- | --- |
| Link length | 500,000 m |
| Wavelength | 1550 nm |
| Rx diameter | 0.05 m |
| Tx diameter | 0.005 m |
| Tx Half-angle diverge | 0.133 mrad |
| Visibility range | 1000 km |
| Pointing jitter | 0.2 mrad |

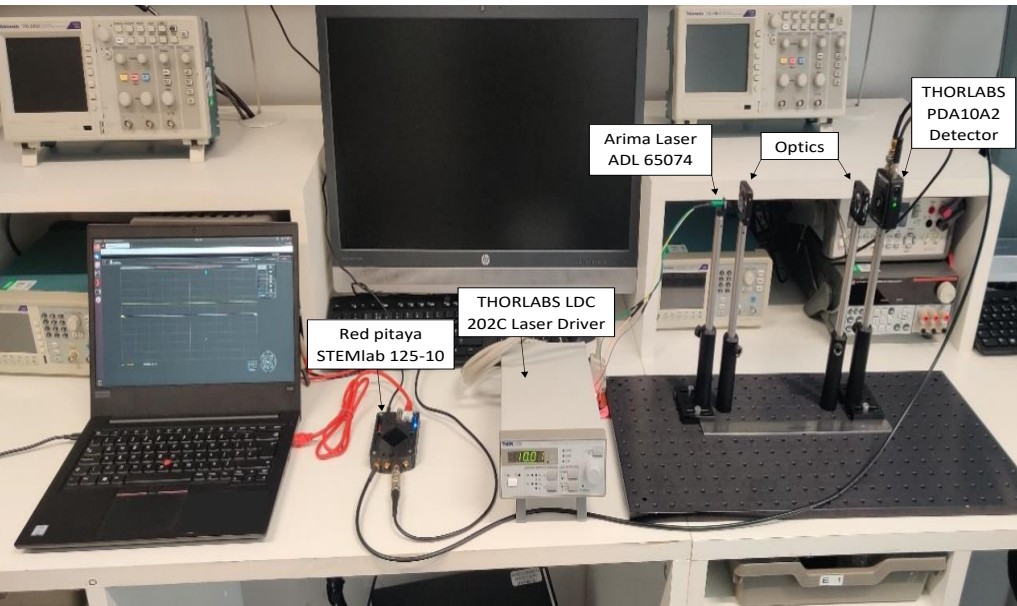

**Figure 10.** Secure FSO lab-based experiment setup.

## 4. Secure FSO Link Experimental Demonstration

Now that a secure FSO link has been demonstrated within a simulation, it can be moved to an FSO link prototype using optics and transceiver hardware. The aim is to replicate the parameters of the simulation in real life and compare the results of the simulation and the physical FSO link. The encryption should hold up both in a simulation and a real environment for it to be considered secure.

### 4.1. Experiment Setup

Figure 10 depicts the lab-based experiment setup of the secure FSO link. A Red Pitaya SDR transceiver acts as the interface between the optical hardware and GNU Radio software in a loop-back connection. The host computer of GNU Radio communicates with the Red Pitaya's onboard FPGA via an Ethernet cable. Once the Red Pitaya processes the data through its digital-to-analogue converter (DAC), it is sent through the output RF port to the optical hardware. The optical hardware consists of a THORLABS LDC 202C laser driver, an Arima laser ADL 65074, 10 mm focal length optics and a THORLABS PDA10A2 detector. The output of the Red Pitaya is connected to the modulation port of the laser driver. The driver modulates the power of the laser to match the output of the Red Pitaya. The laser driver also limits the laser current to 10 mA to prevent any damage. The laser emits visible red light at 650 nm, which is first collimated by an optic and then focused onto the receiver by a second optic of the same specification. By focusing the light correctly, a voltage of 60 mV can be achieved at the optical receiver. The receiver will output a voltage, which can be connected straight into the input port of the Red Pitaya to be processed by its onboard digital down-converter (DDC) and sent, via Ethernet cable, to the GNU Radio interface.

Figure 11 shows the GRC file for the secure FSO link using the SDR and optical hardware. As explained in Section 3, the implementation of the system in GNU Radio consists of four parts: the initial setup, the transmitter, the receiver, and the channel, which is replaced with hardware. The figure shows that the FSK packet transmitter, and receiver are connected to the Red Pitaya sink and source, respectively, using the OOT blocks [31]. The Red Pitaya SDR acts as the interface between the optical hardware and GNU Radio software.

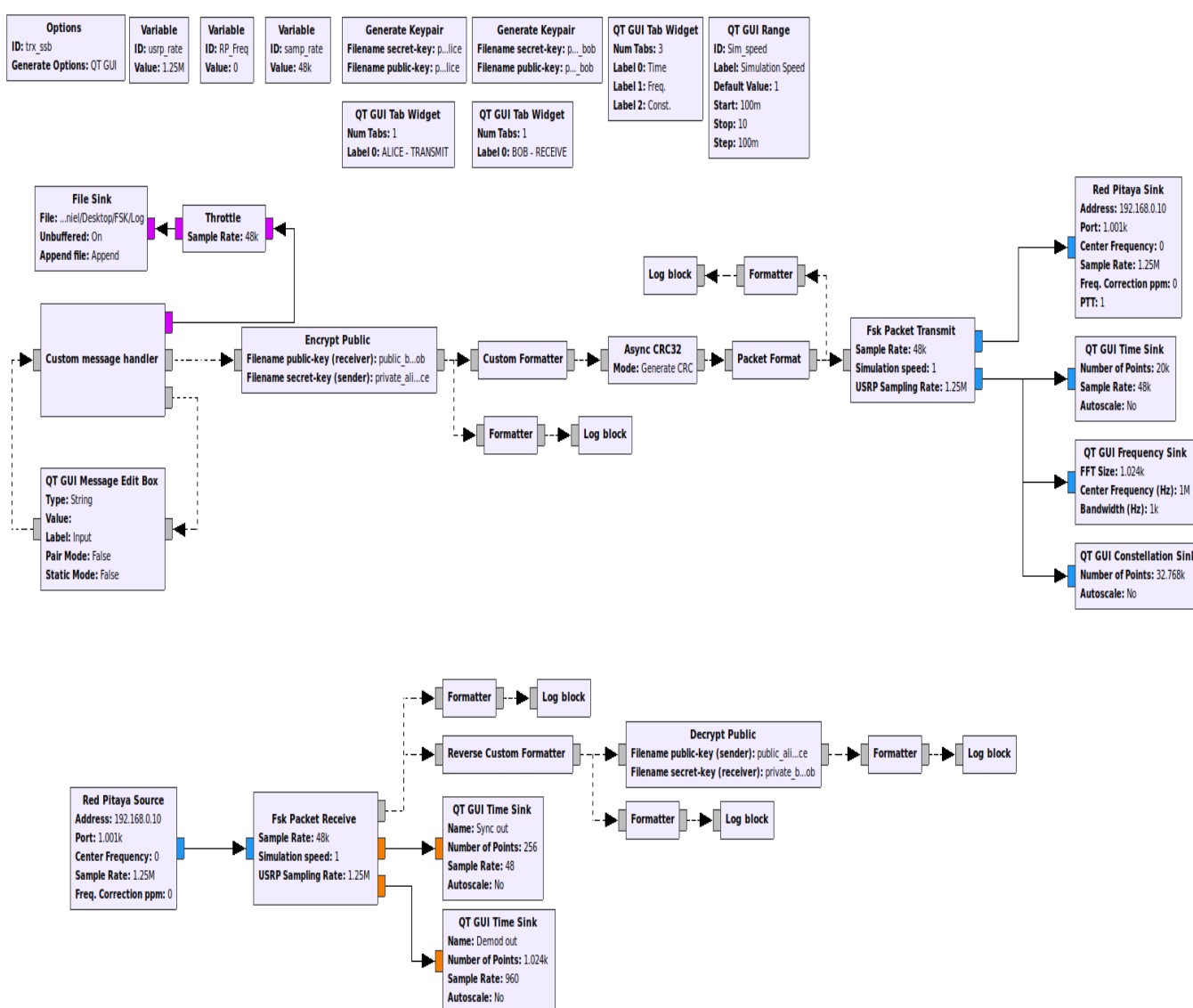

**Figure 11.** The GRC file for the secure FSO link using the optical hardware.

*4.2. Experiment Results*

The experiment tested system resilience against eavesdropping attacks in the presence of Eve. Similar to Section 3.5, Eve generated a private encryption key for the public decryption at the receiver. As shown in Figure 12, the decryption fails even if Eve can intercept the message and decode the header and packet data. This matches the results from the simulation and is the expected result due to the nature of the encryption. It can be concluded that the FSO link is reliable and secure against eavesdropping attacks.

**Figure 12.** FSO link output under eavesdropping attacks.

The experiment results show that a distance of 250 mm between the transmitter and receiver achieves a minimum received electrical amplitude of 27.5 mV required at the Red Pitaya SDR to process the received signal. This achieves a PDR of 92%, as shown in Figure 13a. While Figure 13b shows that reducing the distance to 130 mm increases the received electrical amplitude to 38.5 mV and the PDR to 96%.

Although communication speed is not the driving factor of this experiment, analysis was carried out to find the maximum achievable data rate. The communication speed can be changed by changing the sampling rate of the Red Pitaya, and taking fewer samples, which means a faster overall message speed. When the Red Pitaya samples the transmitter at 1.25 Mb/s, and since each byte sampled is made up of eight bits (from 'init_u8vector' function during message formation), the achievable Red Pitaya transmission rate is 10 Mbit/s.

**Figure 13.** FSO link output at (**a**) 250 mm and (**b**) 130 mm distance.

## 5. Discussion

This study considered a security attack scenario on a non-terrestrial FSO link between a satellite (Alice) and a ground station (Bob). Due to the long link length (e.g., 500 km to 500 million km), the beam radius expands in the range of meters to kilometers [16]. This beam radius expansion allows for the UAV-born passive eavesdropper (Eve) to interrupt the broad optical beam without notifying the legitimate peers Alice and Bob under the hypothesis that Eve is a sufficiently sensitive device that can collect a fraction of leak power $< 10^{-2}$ [5]. Hence, an upper-layer data encryption technique that uses the algorithm (Xsalsa20) within NaCl library was implemented to secure FSO systems. GNU radio simulation platform with SDR hardware was used to prove the effectiveness of the proposed data encryption technique to prevent eavesdropping.

Simulation results of a realistic non-terrestrial FSO link showed that when Eve receives a fraction of leak power of $10^{-2}$ (i.e., 0.1 W of 10 W) and boosts the received signal using an amplifier with a gain of 20 dB, it can decipher the received packets with PDR of 96%. This is consistent with the literature [5] which reported a failure of the physical layer security to prevent eavesdropping at this level of leak power. The results also showed that Eve failed to decrypt any encrypted messages.

The immunity of the cryptographic encryption techniques to eavesdropping attacks was demonstrated experimentally using GNU Radio simulation platform with Red Pitaya SDR and optical hardware. The results also showed that combining encryption with FSO preserved data security.

The results of this study showed that more work is required to secure non-terrestrial optical communication systems. The proposed Xsalsa20 provides sufficient data security against flying eavesdropper that has limited computing power to break the key. However, advanced encryption techniques are required to prevent more powerful malicious security attacks with higher computing capability.

## 6. Conclusions

This study proposed implementing a secured FSO link for non-terrestrial communications against security attacks from flying objects. The system was tested in simulation and experimentally using the GNU Radio platform and software-defined radio hardware. The results proved that implementing cryptographic encryption techniques using the algorithm (Xsalsa20) within NaCl library into FSO systems is effective at stopping eavesdropping attacks and preserving data security. The results also showed that at a distance of 250 mm, the secure system achieved a packet delivery rate of 92% and a transmission rate of 10 Mbit/s. This distance achieves a minimum received electrical amplitude of 27.5 mV required at the receiver SDR to process the data. Combining encryption with FSO helps the adoption of secure non-terrestrial optical communication systems. Xsalsa20 provides sufficient data security against flying eavesdropper that has limited computing power to break the key. More work needs to be done to implement advanced encryption techniques that will increase the versatility of this communication system.

**Author Contributions:** Conceptualization, D.H. and F.M.A.; methodology, D.H. and F.M.A.; software, D.H.; formal analysis, D.H.; investigation, D.H.; resources, D.H. and F.M.A.; data curation, D.H.; writing original draft preparation, D.H. and F.M.A.; writing review and editing, F.B., Z.A., T.S., O.S., O.K.; visualization, D.H.; supervision, F.M.A. All authors have read and agreed to the published version of the manuscript.

**Funding:** This research received no external funding.

**Institutional Review Board Statement:** The study was approved by the Ethics Review and Approval Procedure of Coventry University (Project ID P144609, approved on 30 November 2022).

**Data Availability Statement:** The data presented in this study are available on request from the corresponding author (ad9051@coventry.ac.uk). The data are not publicly available due to intellectual property rights.

**Conflicts of Interest:** The authors declare no conflicts of interest.

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
