# Peer review of "Securing Non-Terrestrial FSO Link with Public Key Encryption against Flying Object Attacks"

_photonics, doi:10.3390/photonics10080884_

Round 1

Reviewer 1 Report

The paper is suitable for publication in the special issue of the journal Photonics. It first shows a simulated communication link with GNU Radio and secondly verifies it in a very basic experimental setup. 

I have the following suggestions:
- In section 3, page 3: Please provide typical sizes and dimensions for FSOC beams in comparison with radio frequency based communication.

- In the section about author contributions: please either provide the full name of author F.S. or correct the typo (F.S.->F.B.)

Author Response

Reviewer 1:

The paper is suitable for publication in the special issue of the journal Photonics. It first shows a simulated communication link with GNU Radio and secondly verifies it in a very basic experimental setup:

Our response: Thank you for your comments. We have responded to individual queries in a point-by-point manner that can be found below.

(1) In section 3, page 3: Please provide typical sizes and dimensions for FSOC beams in comparison with radio frequency-based communication.

Our response: Thank you for your comment. Typical sizes and dimensions for FSOC beams in comparison with radio frequency communications were added in section 3 page 3:

“Figure 1 shows a) schematic and b) geometry of a security attack scenario on a non-terrestrial FSO link between a satellite and ground station as the transmitter (Alice) and receiver (Bob), respectively. The attacker (Eve) is a UAV that interrupted the optical beam. The work in [15] showed that an FSO beam has a spread diameter d ≈ Dθ + R of 50 cm for a divergence angle of θ=0.1 mrad and link length of D =5 km, where R is the beam diameter at the transmitter. In satellite FSO communications, when the link length ranges from 500km for low earth orbit satellites, to 500 million km for deep space optical links, the beam radius expands between 6.63 m to 2.19×105 m [16, P. 164]. This beam expansion is 1000 times less than RF-based satellite communications [17,18] and it can be controlled using optics [19, 20]. However, very narrow FSO beam divergence is not desirable because it causes mis-alignment errors due to satellite vibration or platform jitter [17]. With the advances in UAV technologies, a flying attacker of 10 cm receiver aperture can interrupt the broad optical beam and align to the transmitter which exposes the FSO link to security threats [21], as illustrated in Figure 1 b).”

                                                          (a)                                                                                                                                (b)

Figure 1 Security attack on satellite to ground-station FSO link by a UAV that interrupts the optical beam: a) schematic (not to scale) and b) geometry of the link.

(2) In the section about author contributions: please either provide the full name of author F.S. or correct the typo (F.S.->F.B.).

Our response: Thank you for your comment. To avoid any confusion, full names of authors was provided.

References appeared in this report

  1. Eghbal, M.; Abouei, J. Security enhancement in free-space optics using acousto-optic deflectors. Journal of Optical Communications 413 and Networking 2014, 6, 684–694. https://doi.org/10.1364/JOCN.6.000684.
  2. Barnwell, N. Free-Space Optical Links for Small Spacecraft Navigation, Timing, and Communication. PhD thesis, University of Florida Gainesville, FL, USA, 2018.
  3. Kaushal, H.; Kaddoum, G. Optical Communication in Space: Challenges and Mitigation Techniques. IEEE Communications Surveys and Tutorials 2017, 19, 57–96. https://doi.org/10.1109/COMST.2016.2603518.
  4. Franz, J.; Jain, V.K. Optical communications, components and systems, analysis design optimization application; 2000.
  5. Najafi, M.; Schmauss, B.; Schober, R. Intelligent Reflecting Surfaces for Free Space Optical Communication Systems. IEEE Transactions on Communications 2021, 69, 6134–6151. https://doi.org/10.1109/TCOMM.2021.3084637.
  6. Safi, H.; Dargahi, A.; Cheng, J.; Safari, M. Analytical Channel Model and Link Design Optimization for Ground-to-HAP Free-Space 422 Optical Communications. Journal of Lightwave Technology 2020, 38, 5036–5047. https://doi.org/10.1109/JLT.2020.2997806.
  7. Ortiz, G.G.; Lee, S.; Monacos, S.; Wright, M.; Biswas, A. Design and development of a robust ATP subsystem for the Altair UAV-to-Ground lasercomm 2.5 Gbps demonstration 2003. https://doi.org/2014/7008.

Reviewer 2 Report

The manuscript studies the security issue in FSO links and proposes a security solution for FSO systems. The analysis is performed using simulations and supported by hardware lab implementation. The reviewer has a main concern about this work, which is that the work does not really reflect FSO systems. The simulations and experiments do not show how optical signals can be eavesdropped. When eavesdropping happens, the receiver side can detect such suspicious action by for example monitoring the low received power and declaring an alarm. The work does not discuss how “Eve” can eavesdrop on part of the FSO signal power.

Another concern is about the obtained results. The results do not show any effect of the FSO channel and the FSO system design. What is the effect of the channel? What is the impact of the beam size, link length, etc.? What is the effect of the flying object on the received power? None of the FSO parameters were discussed. The focus was mainly on using the GNU simulator, which is an open-source tool, for simulating the FSO system and implementing security into the system.

The reviewer thinks that the authors must do more analysis and provide solid results that implement the FSO system and answers the raised questions above.

Other comments:    

Many acronyms are not defined such as SDR in line 16, 6G in line 23, VCO in line 192, OOT in line 191. The document should be scanned to find the undefined acronyms.

The word “called” in line 43 is duplicated

Section 2 is very short to be a separate section. It could be part of Section 1.

What is w in equation 1?

What is the size of the UAV, the link length, and the size of the received beam?

In Fig. 5, the tx diameter is 3m. The reviewer feels that this is an unrealistic value. The author should add references from the literature to support their claim.

Author Response

Reviewer: 2

The manuscript studies the security issue in FSO links and proposes a security solution for FSO systems. The analysis is performed using simulations and supported by hardware lab implementation. The reviewer has a main concern about this work:

Our response: Thank you for your comments. We would like to respond to individual queries in a point-by-point manner that can be found below.

(1) The work does not really reflect FSO systems. The simulations and experiments do not show how optical signals can be eavesdropped. When eavesdropping happens, the receiver side can detect such suspicious action by for example monitoring the low received power and declaring an alarm. The work does not discuss how “Eve” can eavesdrop on part of the FSO signal power.

Our response: Thank you for your comment. The manuscript was updated to explain how optical signals can be eavesdropped on when eavesdropping happens and the possibility of attack detection by the legitimate peer Alice and Bob. We also added a new figure to clarify the geometry of the link. The explanation was supported by the outcomes of previous similar studies, as follows:

“Figure 1 shows a) schematic and b) geometry of a security attack scenario on a non-terrestrial FSO link between a satellite and ground station as the transmitter (Alice) and receiver (Bob), respectively. The attacker (Eve) is a UAV that interrupted the optical beam. The work in [15] showed that an FSO beam has a spread diameter d ≈ Dθ + R of 50 cm for a divergence angle of θ=0.1 mrad and link length of D =5 km, where R is the beam diameter at the transmitter. In satellite FSO communications, when the link length ranges from 500km for low earth orbit satellites, to 500 million km for deep space optical links, the beam radius expands between 6.63 m to 2.19×105 m [16, P. 164]. This beam expansion is 1000 times less than RF-based satellite communications [17,18] and it can be controlled using optics [19,20]. However, very narrow FSO beam divergence is not desirable because it causes mis-alignment errors due to satellite vibration or platform jitter [17]. With the advances in UAV technologies, a flying attacker of 10 cm receiver aperture can interrupt the broad optical beam and align to the transmitter which exposes the FSO link to security threats [21], as illustrated in Figure 1 b). In the terrestrial FSO case, Eve was assumed to be a sufficiently sensitive device that can collect a fraction of leak power < 10−2, otherwise, it causes a power reduction that notifies the legitimate peers Alice and Bob [22]. The study in.[22] also showed that relying on the physical layer security of the FSO is not sufficient when a fraction of leak power > 10−2. Hence, an upper-layer data encryption technique is required.”

(a)                                                                                                         (b)

Figure 1 Security attack on satellite to ground-station FSO link by a UAV that interrupts the optical beam: a) schematic (not to scale) and b) geometry of the link.

(2) Another concern is about the obtained results. The results do not show any effect of the FSO channel and the FSO system design. What is the effect of the channel? What is the impact of the beam size, link length, etc.?

Our response: Thank you for your comment. We added new results to the revised manuscript that consider a realistic FSO link parameters calculated in [16] for low earth orbit satellite mission that uses Miniature Optical Communication Transceiver (MOCT), which was developed at the University of Florida. The results were added to section 3.5, page 9 and 11:

“To study the performance of the proposed system under realistic satellite to ground-station beam geometry and channel conditions, a simulation-based FSO link is implemented with realistic parameters shown in Table 1 for low earth orbit satellite mission that uses miniature optical communication transceiver (MOCT) developed at the University of Florida.[16]. Figure 9 a) shows that this FSO link achieves a PDR of 88% using transmitted laser power of 10 W. When the transmitted laser power is reduced to 0.1 W, i.e., a fraction of leakage power 10−2 [22], the PDR decreases to 0%, unless Eve is 100x more sensitive or can boost the received signal using an amplifier with gain of 20 dB. In this case, the PDR increases to 96% as illustrated in Figure.9.b). The simulation results also showed that Eve failed to decrypt any encrypted messages.”

(3) What is the effect of the flying object on the received power?

 Our response:

Thank you for your comment. With the advances in UAV technologies, a flying attacker of 10.cm receiver aperture can interrupt the broad optical beam and align to the transmitter which exposes the FSO link to security threats [21], as illustrated in Figure 1 b). In the terrestrial FSO case, Eve was assumed to be a sufficiently sensitive device that can collect a fraction of leak power < 10−2, otherwise, it causes a power reduction that notifies the legitimate peers Alice and Bob [22]. The study in.[22] also showed that relying on the physical layer security of the FSO is not sufficient when a fraction of leak power > 10−2. Hence, an upper-layer data encryption technique is required. This discussion was added in section 3 page 3:

With the advances in UAV technologies, a flying attacker of 10 cm receiver aperture can interrupt the broad optical beam and align to the transmitter which exposes the FSO link to security threats [21], as illustrated in Figure 1 b). In the terrestrial FSO case, Eve was assumed to be a sufficiently sensitive device that can collect a fraction of leak power < 10−2, otherwise, it causes a power reduction that notifies the legitimate peers Alice and Bob [22]. The study in.[22] also showed that relying on the physical layer security of the FSO is not sufficient when a fraction of leak power > 10−2. Hence, an upper-layer data encryption technique is required.

(4) None of the FSO parameters were discussed.

 Our response: Thank you for your comment. The manuscript provided FSO simulation parameters that replicate a realistic lab-based experiment in section 3.5, the first paragraph. In addition, as the reviewer suggested, we added realistic FSO link parameters calculated in [16] for low earth orbit satellite mission that uses Miniature Optical Communication Transceiver (MOCT), which was developed at the University of Florida. The parameters were summarized in Table 1, page 12:

(5) The focus was mainly on using the GNU simulator, which is an open-source tool, for simulating the FSO system and implementing security into the system.

 Our response: Thank you for your comment. Gnu radio simulation was used as a tool to prove the proposed concept. The proposed encryption scheme is widely applicable to any practical FSO system or simulation platform. The main focus, novelty and original contribution of this study (as highlighted in section 1.1) is to demonstrate a secure FSO system. To the best of our knowledge, none of the previous studies investigated cryptographic encryption in FSO but instead focused on the physical layer security.

(6) The reviewer thinks that the authors must do more analysis and provide solid results that implement the FSO system and answers the raised questions above.

Our response: Thank you for your comment. More results were added to study the effect of the channel and the impact of the beam size, and link length. Please refer to our detailed answers to comments 1 and 2.

(7) Many acronyms are not defined such as SDR in line 16, 6G in line 23, VCO in line 192, OOT in line 191. The document should be scanned to find the undefined acronyms.

Our response: Thank you very much for your comment. The acronyms SDR in line 16, 6G in line 23, VCO in line 192, OOT in line 191 are defined, and the manuscript was proofread for similar errors.

(8) The word “called” in line 43 is duplicated.

 Our response: Thank you, this is corrected.

(9) Section 2 is very short to be a separate section. It could be part of Section 1

 Our response: Thank you for your comments. Section 2 was changed to be subsection 1.1.

(10) What is w in equation 1?

Our response: Thank you for your comment. In equation 1, w is the beam spread and wo is the beam waist. This was stated in line 85 and 86.

(11) What is the size of the UAV, the link length, and the size of the received beam?

Our response: Thank you for your comment. In satellite FSO communications, when the link length ranges from 500 km for low earth orbit satellites, to 500 million km for deep space optical links, the beam radius expands between 6.63 m to 2.19×105 m [16, P. 164].  With the advances in UAV technologies, a flying attacker of 10 cm receiver aperture [21] can interrupt the broad optical beam and align to the transmitter which exposes the FSO link to security threats. This discussion was added to the manuscript section 3 page 3 as mentioned in the first point.  

(12) In Fig. 5, the tx diameter is 3m. The reviewer feels that this is an unrealistic value. The author should add references from the literature to support their claim.

Our response: Thank you for your comment. GNU RADIO shows the SI units prefixes or subunits. For example, in Fig. 5, m stands for milli, u for micro, etc. This means that tx diameter is 3mm and Tx laser power is 275 milliwatts. Fig. 5 was changed to show units.

Figure 5. Secure FSO link Channel blocks in GNU Radio.

References appeared in this report

  1. Eghbal, M.; Abouei, J. Security enhancement in free-space optics using acousto-optic deflectors. Journal of Optical Communications 413 and Networking 2014, 6, 684–694. https://doi.org/10.1364/JOCN.6.000684.
  2. Barnwell, N. Free-Space Optical Links for Small Spacecraft Navigation, Timing, and Communication. PhD thesis, University of Florida Gainesville, FL, USA, 2018.
  3. Kaushal, H.; Kaddoum, G. Optical Communication in Space: Challenges and Mitigation Techniques. IEEE Communications Surveys and Tutorials 2017, 19, 57–96. https://doi.org/10.1109/COMST.2016.2603518.
  4. Franz, J.; Jain, V.K. Optical communications, components and systems, analysis design optimization application; 2000.
  5. Najafi, M.; Schmauss, B.; Schober, R. Intelligent Reflecting Surfaces for Free Space Optical Communication Systems. IEEE Transactions on Communications 2021, 69, 6134–6151. https://doi.org/10.1109/TCOMM.2021.3084637.
  6. Safi, H.; Dargahi, A.; Cheng, J.; Safari, M. Analytical Channel Model and Link Design Optimization for Ground-to-HAP Free-Space 422 Optical Communications. Journal of Lightwave Technology 2020, 38, 5036–5047. https://doi.org/10.1109/JLT.2020.2997806.
  7. Ortiz, G.G.; Lee, S.; Monacos, S.; Wright, M.; Biswas, A. Design and development of a robust ATP subsystem for the Altair UAV-to-Ground lasercomm 2.5 Gbps demonstration 2003. https://doi.org/2014/7008.
  8. Lopez-Martinez, F.J.; Gomez, G.; Garrido-Balsells, J.M. Physical-Layer Security in Free-Space Optical Communications. IEEE Photonics Journal 2015, 7, 1–14. https://doi.org/10.1109/JPHOT.2015.2402158.

Round 2

Reviewer 2 Report

The reviewer is satisfied with the authors' responses